# Unraveling the Resistance of IGF-Pathway Inhibition in Ewing Sarcoma

**DOI:** 10.3390/cancers12123568

**Published:** 2020-11-29

**Authors:** Stefanie de Groot, Bas Röttgering, Hans Gelderblom, Hanno Pijl, Karoly Szuhai, Judith R. Kroep

**Affiliations:** 1Department of Medical Oncology, Leiden University Medical Center, P.O. Box 9600, 2300 RC Leiden, The Netherlands; s.de_groot2@lumc.nl (S.d.G.); a.j.gelderblom@lumc.nl (H.G.); 2Department of Cell and Chemical Biology, Leiden University Medical Center, P.O. Box 9600, 2300 RC Leiden, The Netherlands; Bas1.rottgering@live.nl; 3Department of Endocrinology, Leiden University Medical Center, P.O. Box 9600, 2300 RC Leiden, The Netherlands; H.Pijl@lumc.nl

**Keywords:** IGF1, IGF2, IGF1R, insulin, INSR, IRA, IRB, GH, resistance, Ewing sarcoma

## Abstract

**Simple Summary:**

The insulin-like growth factor-1 receptor (IGF1R) is a receptor commonly overexpressed and overactivated in a variety of cancers, including Ewing sarcoma, and promotes cell growth and survival. After promising results with targeting and inhibiting the receptor in vitro, multiple different IGF1R targeting compounds have been clinically tried but showed limited efficacy. Here we discuss several possible resistance mechanisms which could explain why IGF1R targeting fails in the clinic and discuss possible ways to overcome these resistances.

**Abstract:**

Insulin-like growth factor-1 receptor (IGF1R) inhibitors are effective in preclinical studies, but so far, no convincing benefit in clinical studies has been observed, except in some rare cases of sustained response in Ewing sarcoma patients. The mechanism of resistance is unknown, but several hypotheses are proposed. In this review, multiple possible mechanisms of resistance to IGF-targeted therapies are discussed, including activated insulin signaling, pituitary-driven feedback loops through growth hormone (GH) secretion and autocrine loops. Additionally, the outcomes of clinical trials of IGF1-targeted therapies are discussed, as well as strategies to overcome the possible resistance mechanisms. In conclusion, lowering the plasma insulin levels or blocking its activity could provide an additional target in cancer therapy in combination with IGF1 inhibition. Furthermore, because Ewing sarcoma cells predominantly express the insulin receptor A (IRA) and healthy tissue insulin receptor B (IRB), it may be possible to synthesize a specific IRA inhibitor.

## 1. Introduction

Insulin-like growth factor-1 receptor (IGF1R) and other members of the IGF1 receptor pathway have been associated with the development, progression and metastasis of cancer and resistance to anticancer therapies [1]. Convincing preclinical evidence supporting the efficacy of IGF1R pathway inhibition in the treatment of cancer has led to the development of many IGF1R pathway inhibitors [2,3,4,5,6], which have been investigated in numerous clinical trials in breast cancer; Ewing sarcoma and various other types of solid tumors, like non-small cell lung cancer, hepatocellular, gastric and esophageal carcinoma (Table 1) [7,8,9,10,11,12,13,14,15,16,17,18,19,20,21,22,23,24,25,26]. Unfortunately, no convincing benefit of IGF1R pathway inhibitors has been found in these studies [27,28], except in some rare cases of a sustained response in patients with Ewing sarcoma and adrenocortical carcinoma [9,29]. In Ewing sarcoma, IGF1R was early identified as a possible target for the treatment, of which the preclinical results were published, where Scotlandi et al. showed the proliferative effect of IGF1 on Ewing sarcoma cell lines [30]. However, later clinical trials exhibited disappointing results due to therapy resistance to IGF1R inhibition (Table 1).

Possible resistance mechanisms that could explain the disappointing results of IGF1R inhibitors in the clinical setting are: (1) inadequate inhibition of the pathway downstream of IGF1R, as this pathway can be activated through the IGF1R but, also, through the insulin receptor A (IRA) or hybrids between the two receptors [31,32,33]; (2) the disruption of negative feedback loops in the pituitary, whereby the IGF1R ligands IGF1, IGF2 and insulin, and other endocrine-signaling molecules such as the growth hormone (GH) and glucose, increase by inhibiting the receptors, leading to tumor growth [34,35]; (3) the existence of autocrine or paracrine feedback loops in the tumor, through which the IGF1R pathway is continuously activated, perhaps via intracellular routes [11,36] and (4) tumor growth and survival due to other driver mutations downstream of the IGF1R pathway or in other oncogenic pathways, which makes inhibiting IGF1R irrelevant [37,38].

This review covers the current knowledge of IGF1R pathway inhibitors and their possible resistance mechanisms that may explain the disappointing results of IGF1R pathway inhibitors in the clinical setting. We also discuss potential strategies to overcome the resistance mechanism to guide future IGF1R inhibitor research and therapy.

## 2. IGF1/Insulin Pathway and Cancer

The IGF1/insulin pathway regulates growth in normal tissues and is associated with cancer development and reduced cancer survival rates. The pathway has been extensively described before [6,39]. Briefly, IGF1R, IRA or hybrid receptors can be activated by binding IGF1, IGF2 and insulin ligands, which leads to activation of the RAS/MAPK and PI3K/AKT downstream pathways (Figure 1) [40,41]. Each ligand has a specific affinity for each receptor [42,43]. The IGF1R and IRA are both frequently overexpressed in distinct types of cancers, including breast, colorectal and prostate carcinoma [43]. Additionally, although recurrent activating mutations in the IGF1R are unknown, single-nucleotide polymorphisms with unknown significance have been described [44].

Epidemiological studies have shown a relationship between high circulating IGF1 levels and cancer incidence [45,46]. Counterintuitively, high levels of IGF1 in the plasma and cytoplasm of the cancer cells seem to be a prognostic for the improved survival in cancer patients [11,16,47,48]. High baseline IGF1 was counterintuitively associated with improved event-free survival (EFS) in Ewing sarcoma patients. Additionally, in Ewing sarcoma, patients with metastatic disease exhibited lower IGF1 levels when compared to the localized disease, suggesting that further progression of the disease negatively modulates IGF1 levels, which would explain the higher EFS in patients with higher IGF1 levels [47]. However, during systemic treatment, increases of the levels of IGF1 seem to not be predictive for the treatment outcome [48,49].

### 2.1. Insulin Receptor and Insulin

The IR, also known as INSR, has two different isoforms: IRA and insulin receptor B (IRB) [43]. The difference between these two receptor variants is that IRA is 12 amino acids shorter than IRB, due to the alternative splicing of exon 11 [43]. This difference results in a distinct affinity for their ligands. Insulin and IGF2 can bind the IRA, while IRB only binds insulin with high affinity, whereby different downstream pathways are activated: IRB fulfills an important role in glucose homeostasis, whereas IRA, the embryological splice variant, is the dominantly expressed isoform in many cancer cells [31]. The IRA activates proliferation and antiapoptotic pathways, and its expression is associated with resistance to cancer treatment [50]. Interestingly, reduced insulin levels as a result of a very low-caloric diet and weight loss are associated with a relative IRB mRNA increase, without affecting the total gene expression of IR in adipose tissue [51]. Hyperinsulinemia, in the context of obesity, insulin resistance or type 2 diabetes mellitus, is also associated with an increased risk of cancer and cancer therapy resistance [52,53].

### 2.2. IGF2R and IGF Binding Proteins

IGF2 can bind to the IGF1R and IRA, but it can also bind to the IGF2 receptor (IGF2R), which is also known as the Cation-Independent Mannose-6-Phosphate Receptor. This receptor is considered a tumor suppressor, as both the ligand and receptor are internalized and degraded after binding, reducing the bioavailability of IGF2 and, thus, inhibiting the proliferative effects of IGF2 [54,55]. Moreover, six IGF-binding proteins (IGFBPs) exist, which can bind and inactivate IGF1 and IGF2 by blocking them from binding to their receptor while protecting them from degradation and increasing their half-life [56,57]. Interestingly, these binding proteins simultaneously enhance IGF signaling locally and increase IGF availability for eventual binding to the IGF1R [57]. Some data suggest that higher levels of IGFBP3, the main IGF-binding protein, are associated with an increased risk of cancer [46], while others support an inverse association [47,58]. For example, a significant correlation between increased IGFBP mRNA expression in tumor tissues and increased patient survival has been reported. Furthermore, an increase in IGF1R signaling in response to IGFBP3 downregulation has been indicated as a possible resistance mechanism in cancer treatment [59]. Therefore, IGFBPs may be a possible tumor suppressor in tumors with active IGF1R signaling [59]. Apart from the endocrinal function, IGFBPs also have functions in a variety of other processes. IGFBPs can bind to cell-surface receptors and internalize into the cell. After internalization, the IGFBPs can induce apoptosis and change transcriptional regulation [59]. However, the exact role of the IGFBPs in different cancers remains unclear and requires more study.

## 3. IGF1R Pathway Inhibitors and Resistance

IGF1R pathway inhibitors possess different properties to inhibit the IGF pathway and can be classified into three groups: (1) IGF1R antagonist monoclonal antibodies, (2) IGF tyrosine kinase inhibitor small molecules and (3) IGF ligand neutralizing antibodies [6,39].

### 3.1. IGF1R Antagonist Monoclonal Antibodies

IGF1R antagonist monoclonal antibodies bind selectively to IGF1R with high affinity and block the interaction of IGF1R with its ligands, inducing the internalization and degradation of IGF1R [5,60]. The IGF1R antagonist monoclonal antibodies ganitumab (AMG-479) [13], dalotuzumab (MK-0646) [27], cixutumumab (IMC-A12) [12], teprotumumab (R-1507) [16] and figitumumab (CP-751871) were tested in clinical studies (Table 1) [61]. These inhibitors induced downregulation of the IGF1R homodimers and hybrid receptors (e.g., IGF1R/IRA), while the integrity of IRA homodimers and their activation by insulin or IGF2 was not influenced [61]. Indeed, Schmitz et al. found decreased IGF1R expression in patients treated with figitumumab but the absence of the inhibition of AKT, leading to the hypothesis that the downstream pathway remains activated [61]. In clinical studies, figitumumab and other IGF1R antibody antagonists were shown to increase circulating IGF1 and growth hormone (GH) levels, as well as glucose and insulin plasma levels [11,23,61]. Thus, the activation of downstream pathways and an increase of several growth factors despite the IGF1R blockade may explain the failure of these compounds in the clinical setting. Patients whose tumors express IGF1R but not IRA may, however, benefit from IGF1R inhibitors, which might explain why these compounds caused long-lasting tumor response in two cases in clinical trials (Table 1) [9].

### 3.2. IGF1R Tyrosine Kinase Inhibitor Small Molecules

IGF tyrosine kinase inhibitor small molecules, such as linsitinib (OSI-906), BMS-754807 and KW-2450, target both the IGF1R and the insulin receptor (IR) and their hybrid receptors [18,62]. Puzanov et al. found that linsitinib decreased phosphorylation of the IGF1R and IR in peripheral blood mononuclear cells (PBMC) [18]. Accordingly, hyperglycemia and hyperinsulinemia are common side effects of these agents due to cross-reactivity with the insulin receptor B (IRB), which is involved in glucose metabolism [18]. This can lead to the discontinuation of treatment and may also cause resistance to this kind of inhibitor. The IGF tyrosine kinase inhibitor small molecules did not show a survival benefit in advanced or metastatic adrenocortical carcinoma in a large phase 3 trial [20] and showed disappointing results in other (small) clinical trials (Table 1) [18,19,21,22].

### 3.3. IGF Ligand Neutralizing Antibodies

The IGF ligand neutralizing antibodies dusigitumab (MEDI-573) and xentuzumab (BI 836845) inhibit the IGF1R and the IRA by binding and neutralizing both IGF1 and IGF2 ligands [23]. In contrast with the IGF1R antagonist monoclonal antibodies and IGF1R tyrosine kinase inhibitors, these compounds do not cause hyperglycemia, as they do not compromise insulin action [23,24]. However, as insulin can also activate the IRA and hybrid receptors, the IGF1R pathway may not be adequately inhibited. Only a few small clinical trials in heavily pretreated patients have been performed to date that showed only a few partial responses (Table 1) [9,24,25,29,63].

## 4. Strategies to Overcome Resistance Mechanisms of IGF1R-Inhibitors

Several strategies have been proposed to overcome the mechanisms of resistance to the different IGF1R inhibitor types.

### 4.1. Activation of IRA and/or Hybrid Receptors

As previously described, the pathway downstream of the IGF1R may be inadequately inhibited by all three distinct inhibitors. While the IGF1R itself is appropriately inhibited, the IRA and its hybrid receptors may still be activated by its ligands (IGF1, IGF2 and/or insulin) [31,32,33]. This indicates that signaling through the IRA may be an important resistance mechanism to anti-IGF1R treatment. In support of this, it has been shown that IGF1R inhibition can lead to compensatory IR activation in colorectal cancer, ovarian carcinoma, and Ewing sarcoma in vitro [64]. The addition of a specific IRA inhibitor would be required to overcome this, as nonspecific IR inhibitors (e.g., the compound S961) and IGF1R tyrosine kinase inhibitor small molecules cause hyperglycemia and compensatory hyperinsulinemia [18,65]. However, specific IRA inhibitors are not yet available. Developing a specific IRA antagonist may serve as a novel treatment option combined with IGF1 inhibitors, as this may be an option with knowledge about the crystal structure [66]. Alternatively, (short-term) fasting during treatment with an IGF1R inhibitor may have similar effects, as it causes a significant decrease in insulin serum levels [67,68]. Longer periods of dietary restriction are required to significantly reduce IGF2 levels [67], which could still activate downstream pathways through IRA activation. Therefore, more studies are needed to evaluate the efficacy of (short-term) fasting as an adjunct to IGF1R treatment in patients with cancer [68,69,70].

### 4.2. Disruption of Negative Feedback

Another mechanism of resistance to IGF1R pathway inhibition in solid tumors is the increase of plasma GH due to the lack of negative feedback by IGF1 both in the pituitary and hypothalamus, which enables a higher release of GH [71]. This phenomenon is seen in clinical trials with IGF1R antagonist monoclonal antibodies and IGF1R tyrosine kinase inhibitor small molecules, which may blunt the efficacy of these drugs. Additionally, independent potentiating effects of GH that are not mediated by IGF1 have been demonstrated on breast cancer cells [72,73,74]. For example, GH induces tumor growth without increasing IGF1 [74]. This is supported by the fact that several cancers express GH receptors (GHR), and GHR positivity is predictive of a worse outcome [75,76,77,78]. However, our preclinical data did not show a stimulatory effect of GH on Ewing sarcoma cells in vitro (Appendix A, Figure A1). Additionally, GH diminishes the anti-IGF1R tumor inhibition activity, suggesting that increased GH is a plausible cause of IGF1R inhibitor failure in the clinic [74]. Another preclinical study showed that GH causes chemoresistance despite the presence of an IGF1R antagonist monoclonal antibody. In this study, the cancer cells became chemosensitive again in the presence of the GH antagonist pegvisomant [79]. Increased GH levels cause increased IGF1 levels, hyperinsulinemia, insulin resistance and, ultimately, hyperglycemia [35]. Accordingly, patients with acromegaly, who have high GH plasma levels, show a higher incidence of cancer [80,81], while patients with Laron syndrome who are resistant to GH due to a defective GHR and patients with GH deficiency have reduced cancer susceptibility [82,83].

It is proposed that high levels of IGF1 cause resistance to IGF1R inhibitors due to a competitive affinity to the IGF1R receptor. For example, an excess of IGF1 reverses the inhibitory effect of figitumumab in preclinical studies, which is presumed to be due to their similar affinity for the IGF1R [84]. A solution to overcome this may be to increase the dose of the IGF1R inhibitor or to decrease the IGF1 serum levels by adding a GH antagonist, such as pegvisomant [35,85]. In a clinical phase I study (NCT00976508), two patients with Ewing sarcoma had partial responses to the combination treatment of figitumumab and pegvisomant [63]. Unfortunately, the study was stopped prematurely due to the cessation of figitumumab production.

Furthermore, increased insulin secretion activates the IR and may explain the suboptimal therapeutic benefits. Again, an IRA inhibitor and/or short-term fasting in combination with IGF1 inhibitors may be an effective approach to decrease insulin signaling and adequately inhibit the downstream pathway. Our preclinical data supports that insulin clearly stimulates cell growth and blocks the apoptosis (Appendix A, Figure A1) of Ewing sarcoma cells in vitro. Additionally, stimulation with insulin reversed an increase in PARP cleavage, a marker for apoptosis, induced by IGF1R blocking. Furthermore, stimulation with insulin increased AKT phosphorylation in cells treated with an IGF1R inhibitor. This indicates that lowering insulin levels or blocking the IRA may increase the efficacy of IGF1R inhibitors. Additionally, IGF1R blocking can induce hyperglycemia and hyperinsulinemia in patients [7], which could activate the IRA in response to IGF1R inhibition. However, the exact mechanism for the observed hyperglycemia as a side effect of IGF1 inhibitors is unclear, but cross-reactivity with the IRB, which is involved in the glucose metabolism, is likely to be an important factor [18].

### 4.3. Autocrine Loops in the Tumor

Autocrine activation by the tumor is described in preclinical studies, whereby both the IGF1R and one of its ligands are expressed by the tumor or surrounding tumor stroma [11,36]. This would continuously activate the IGF1R pathway, perhaps even via an intracellular route, making it impossible to inhibit with an antibody-based approach. In line with this, lowering the serum GH and/or IGF1 by somatostatin analogs does not always have antitumor effects in clinical studies in breast cancer [75].

### 4.4. Activation or Mutation of Other Pathways

Finally, resistance may occur when the IGF1R pathway is activated through downstream mutations of the pathway (such as PTEN) or in bypassing oncogenic pathways (such as epidermal growth factor receptor (EGFR)) [37,38].

To overcome these resistance mechanisms, it may be necessary to utilize combination therapies to simultaneously block all pathways contributing to tumor growth [39,86]. Combination therapies with IGF1 inhibitors are extensively reviewed elsewhere [39,86]. IGF ligand neutralizing antibodies are good candidates due to the lack of side effects, such as hyperinsulinemia and hyperglycemia, and may be combined with EGFR family inhibitors, Cyclin-dependent kinase (CDK) inhibitors, endocrine therapy or immune checkpoint inhibitors [39].

### 4.5. Use of Biomarkers

If IGF1R is not (overly) expressed by the tumor, it is probably not meaningful to use IGF inhibitors, as the pathway is probably not involved in tumor genesis, growth and therapy resistance. In these cases, it is necessary to determine biomarkers, such as secondary mutations, receptor levels and isoform identification of the IR to select patients who may benefit from treatment. It is particularly important to make use of biomarkers such as the expression levels of IGF1 and IGF2 in tumors with autocrine loops to predict if a patient will benefit from treatment with an IGF inhibitor.

The described resistance mechanisms and potential strategies to combat them are summarized in Table 2 and Figure 1.

### 4.6. Ewing Sarcoma vs. Other Solid Tumors

Ewing sarcoma is a rare cancer [87] that is characterized by a translocation that increases the bioactivity of IGF1 [47,88]. In 85% of cases, the somatic translocation t(11;22) results in the aberrant product of the Ewing sarcoma breakpoint region 1 (EWSR1) gene and Friend leukemia virus integration 1 (FLI1) gene [89] and other variants of the involved gene families in the remaining cases [90]. The product is the EWSR1-FLI1 fusion protein, which binds—amongst other things—to the IGFBP3 promoter, which leads to a dramatic reduction in the expression of IGFBP3 [47,58,88] without inhibiting the availability of IGF1 [91,92,93]. IGF1R was early identified as a target in Ewing sarcoma, as the IGF1R was highly expressed in Ewing sarcoma cell lines in addition to the expression of IGF1, which may thus signal in an autocrine loop [30]. Additionally, the IGF1R inhibition experiments reduced the growth of Ewing sarcoma both in vitro [94] and in vivo [95]. However, in clinical trials, IGF1R inhibitory compounds have not shown the same efficacy (Table 1). Nonetheless, a few patients with Ewing sarcoma experienced a long-term response to IGF1R inhibitor therapy [9,29]. It is not clear why only these few patients showed a clinical benefit. The activation of the IRA may cause resistance to specific IGF1R inhibitors as resistant cells switch from IGF1-IGF1R signaling to IGF2/insulin/IRA signaling, activating the same proliferative downstream pathways [31]. This may indicate that these responding patients with Ewing sarcoma did not have active IRA signaling. However, there is no data to support this, but it should be investigated further. Additionally, a meta-analysis of five clinical trials by Amin et al. showed a potential synergistic effect of mechanistic Target of Rapamycin (mTOR) inhibitors and IGF1R monoclonal antibodies in Ewing sarcoma patients [96]. Since mTOR signaling is a downstream target of both the IGF1R and the IRA, inhibiting mTOR might indeed be a viable treatment option, in addition to IGF1R inhibition. Therefore, the lack of a response in patients with Ewing sarcoma may reflect alterations in pathways that are not disrupted by IGF1R inhibition and/or the other resistance mechanisms mentioned above. Garofalo et al. [97] identified multiple functional pathways associated with IGF1R inhibition resistance. Of the pathways identified, the MAPK kinase pathway and, again, the IGF2/insulin/IRA pathways seem to be important for the resistance to IGF1R inhibition, in addition to a variety of other pathways. Furthermore, the IGF2 and IRA expression increased in vitro in response to IGF1R inhibition with figitumumab [97]. Together, this indicates that, for a better efficacy of IGF1R inhibition in the clinic, either better IRA inhibition is needed and a better understanding of other pathways involved in resistance to IGF1R inhibition like mTOR or the pathways outlined by Garofalo et al. are required [97]. Through this, we can begin to better understand the pathways that could be co-targeted in conjunction with IGF1R inhibition to avoid the IGF1R inhibition resistance. Additionally, the role of the IGF2 mRNA-binding protein 3 (IGF2BP3) in IGF1R and IRA signaling needs to be better understood. In Ewing sarcoma, this oncofetal protein can mediate IGF1R loss and subsequent compensatory IRA and IGF2 activation in some cell lines [98]. In line with this, cell lines with a decreased expression of IGF2BP3 exhibited a higher sensitivity to OSI-906, which means that IGF2BP3 could be a biomarker for IGF1R inhibition [98].

## 5. Discussion

In this review, we summarized several hypotheses of mechanisms of resistance that may explain the disappointing results of IGF1R pathway inhibitors in clinical studies.

First, in the clinical setting, IGF1R inhibition with IGF1R antagonist monoclonal antibodies or IGF tyrosine kinase inhibitor small molecules causes hyperglycemia and subsequent hyperinsulinemia due to cross-reactivity with the IRB and hybrid receptors [20,35]. Therefore, activation of the IGF1/insulin pathway through insulin could be an important resistance mechanism that prevents IGF1 inhibition from achieving clinical efficacy. This indicates that lowering the plasma insulin levels or blocking its activity could provide an additional target in cancer therapy and may be effective in combination with IGF1 inhibition. This is supported by our data (Appendix A, Figure A2), which showed that insulin reverts the inhibitory effect of OSI-906 on Ewing sarcoma cells in vitro. Short-term fasting may also be a valuable addition to IGF1R inhibition, as it dramatically lowers the insulin and IGF1 [68,70,99].

Second, as the IRA is expressed in Ewing sarcoma cell lines and other solid tumors [43], blocking the IGF1R alone could be insufficient to achieve clinical benefit. Therefore, IGF1 inhibition with a receptor antagonist or a tyrosine kinase inhibitor could be combined with an IR inhibitor [100]]. A specific IRA inhibitor would be an optimal addition to the IGF tyrosine kinase inhibitor small molecules to prevent metabolic side effects caused by inhibiting IRB and the subsequent therapy resistance. Given the Ewing sarcoma cell lines predominantly express the IRA variant, and the 12 amino acid differences in the extracellular domains of IRA and IRB [43,50], the specific inhibition of IRA may in itself be an effective treatment of Ewing sarcoma.

Third, it was postulated that an increase of GH through an inhibited feedback loop by blocking IGF1 signaling might induce cell growth and resistance to IGF1 inhibition. However, our results suggest that GH has no effect on Ewing sarcoma cells in vitro (Appendix A, Figure A1). A combination treatment of IGF1R inhibition with pegvisomant, a growth hormone receptor antagonist, has been tried in a phase 1 trial [63], but the final results are not published yet.

Finally, we propose that autocrine loops and other secondary mutations could be the reason for the failure of IGF1R inhibitors in Ewing sarcoma and other solid tumors. Therefore, it is necessary to measure biomarkers such as universal secondary mutations (e.g., TP53, STAG2, IGF2BP3 and the CDKN2A/CDKN2B status in Ewing sarcoma patients) [101]; IGF1 and IGF2 ligand levels and IGF1R, IR and IRA receptor expression in select patients who may benefit from treatment with IGF1R inhibitors. In addition, it may be possible to personalize the treatment with combined treatment strategies based on these biomarkers [39]. Tumors of patients included in phase I trials may be resistant to IGF1R inhibition treatment due to secondary mutations caused by (extensive) pretreatment, and IGF1R inhibition might be more effective as a first-line treatment. However, driver mutations are still positive in 92% of pretreated patients with different tumor types [102].

## 6. Conclusions

The failure of IGF1R inhibitors in clinical studies may be caused by resistant tumors due to secondary mutations in pretreated patients. The complexity of the IGF1R pathway may also play a role in their failure, as pathway activation may not be adequately inhibited due to the insulin and IGF2 activation of IRA, as supported by our preclinical data. Future research should aim to assess the efficacy of combination treatments utilizing IGF1R inhibition and IRA inhibition, lowering insulin and the use of personalized treatments based on tumor biomarkers and ligand levels in patients with solid tumors and, in particular, in patients with Ewing sarcoma.

## Figures and Tables

**Figure 1 cancers-12-03568-f001:**
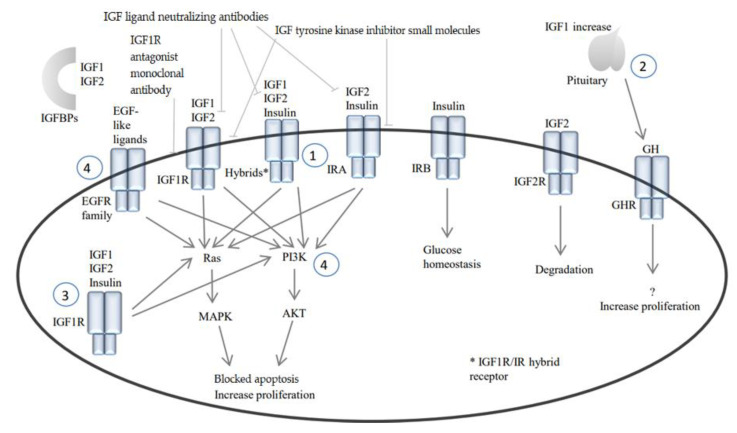
IGF and insulin signaling. Receptor hybridization, activation and downstream signaling of IGF, insulin and GH receptors. IGF1/2 = insulin-like growth factor-1/2, IGF1R = insulin-like growth factor-1 receptor, IGF2R = insulin-like growth factor 2 receptor or mannose 6-phosphate receptor, IR(A/B) = insulin receptor (A/B), GH = growth hormone, GHR = growth hormone receptor, EGF = epidermal growth factor and EGFR = epidermal growth factor receptor.

**Table 1 cancers-12-03568-t001:** IGF1 receptor (IGF1R) pathway inhibitor monotherapy in clinical studies involving patients with Ewing sarcoma and other types of solid tumors.

Study	Trial	Patients	Compound	Endocrine Side Effects and Biomarkers	Clinical Response
IGF1R Antagonist Monoclonal Antibodies					
Haluska et al., 2007, [7]	Phase I	24 patients with distinct solid tumors or sarcoma	Figitumumab (CP-751, 871)	-Hyperglycemia, increase of insulin, GH and IGF-1	7/12 SD, 1 long responder
Tolcher et al., 2009, [8]	Phase I	53 patients with distinct tumors and sarcoma	Ganitumab (AMG 479)	-5 patients with hyperglycemia -IGF1 levels increase during treatment -patients with complete response possess IGF1R in metastases	1 CR, 2 PR
Olmos et al., 2010, [9]	Phase I	29 patients with distinct sarcoma (16 Ewing sarcoma)	Figitumumab (CP-751, 871)	-5 patients with hyperglycemia	1 CR, 1 PR (both Ewing sarcoma), 8 SD
Kurzrock et al., 2010, [10]	Phase I	35 patients with distinct solid tumors or sarcoma (9 Ewing)	Teprotumumab (R1507, RO4858696)	-2 patients with hyperglycemia -IGF1 serum levels increase during treatment	2/33 PR, 14/33 SD
Juergens et al., 2011, [11]]	Phase I/II	31 (phase 1) and 107 (phase 2) patients with distinct sarcoma (16 and 107 Ewing, respectively)	Figitumumab (CP-751, 871)	-3 patients with grade 3 hyperglycemia -IGF1 baseline levels were prognostic for survival, higher levels were associated with better survival -Highest IGF1 level showed a reduced clinical benefit -Increase of serum levels of IGF1, GH and insulin during treatment	15/106 PR, 25/106 SD
Malempati et al., 2012, [12]	Phase I/II	47 patients with distinct solid tumors or sarcoma (35 Ewing)	Cixutumumab (IMC-A12)	-14/44 patients hyperglycemia -Increase in serum levels IGF-I and IGFBP-3 -No change in serum levels IGF-II and IGFBP-2	3/25 PR, 5/25 SD (Ewing sarcoma), 2/13 SD (Other)
Murakami et al., 2012, [13]	Phase I	19 patients with distinct solid tumors	Ganitumab (AMG 479)	-IGF1 and IGFBP3 increased after administration, GH not -IGF1 and IGFBP3 were not predictive or prognostic for a response of treatment	7/19 SD
Tap et al., 2012, [14]	Phase II	38 patients with distinct sarcoma (22 Ewing sarcoma)	Ganitumab (AMG 479)	-5/38 hyperglycemia (2 pts grade III) -IGF1 serum level increased	2/35 PR, 21/35 SD
Schöffski et al., 2013, [15]	Phase II	113 patients with distinct sarcoma (18 Ewing sarcoma)	Cixutumumab (IMC-A12)	-22/111 hyperglycemia (6 patients, grade III)	2/111 PR, 44/111 SD
Pappo et al., 2014, [16]	Phase II	163 patients with distinct sarcoma	R1507	-15/163 hyperglycemia (4 patients, grade III)	4/163 PR, 42/163 SD
Abou-Alfa et al., 2014, [17]	Phase II	24 patients with hepatocellular carcinoma	Cixutumumab (IMC-A12)	-24/24 hyperglycemia (11 patients, grade III)	7/24 SD
Frappaz et al., 2016, [26]	Phase II	20 patients with distinct sarcoma (6 Ewing sarcoma)	Dalotuzumab(Mk-0646)	- 3/20 hyperglycemia	1 PR
IGF1R/IR Dual Inhibitors					
Puzanov et al., 2014, [18]	Phase I	95 patients with distinct solid tumors and sarcoma	Linsitinib (OSI-906)	-4 patients with hyperglycemia -Efficacy independent of KRAS mutation -Increase of IGF1 serum levels	30/95 SD
Jones et al., 2015, [19]	Phase I	97 patients with distinct solid tumors and sarcoma	Linsitinib (OSI-906)	- 37% hyperglycemia -Increase of IGF1 serum levels	2/66 PR, 27/66 SD
Fassnacht et al., 2015, [20]	Phase III	90 patients with adrenocortical carcinoma	Linsitinib (OSI-906)	-2 patients with grade III hyperglycemia -IGF1 serum levels increase	3/90 PR
Barata et al., 2018, [21]	Phase II	17 patients with metastatic castrate-resistant prostate cancer	Linsitinib (OSI-906)	-8 patients with hyperglycemia	1/17 PR, 8/17 SD
Chiappori et al., 2016, [22]	Phase II	29 patients with small cell lung cancer	Linsitinib (OSI-906)	-7/29 hyperglycemia (1 patient grade III)	1/29 SD
IGF1/2 Neutralizing Antibody					
Haluska et al., 2014, [23]	Phase I	43 patients with distinct solid tumors (1 Ewing)	Medi-573	-1 Patient with hyperglycemia -No elevation of insulin or GH -IGF1 and IGF2 suppressed	13/39 SD
Iguchi et al., 2015, [24]	Phase I	10 patients with distinct solid tumors	Medi-573	-1 patient with hyperglycemia -IGF1/2 decreased	4/10 SD
De Bono et al., 2020, [25]	Phase I	125 patients with distinct solid tumors and sarcoma	Xentuzumab	-2 patients with grade III hyperglycemia -IGF bioactivity decreased, total levels did not decrease -No effects on IGF2	2/125 PR, 55 SD

SD: stable disease, PR: partial response and CR: complete response. IGF1/2: Insulin-like growth factor-1 and 2. GH: growth hormone.

**Table 2 cancers-12-03568-t002:** Resistance mechanisms.

Resistance Mechanism	Example	Proposed Solution
Activation of the pathway trough IRA or hybrid receptors	IGF1R is inhibited, but IRA and hybrids receptors still activate the downstream pathway	Add an IRA inhibitor
		Short-term fasting
Abrogation of negative feedback	High levels of IGF1 still activate the receptor due to a competitive affinity	Increase dose of IGF1-inhibitor
		Decrease IGF1 levels by adding GH antagonist [35,79]
	High levels of insulin activate IRA and hybrid receptors	Add an IRA inhibitor
		Short-term fasting
	High levels of glucose	Short-term fasting
	High levels of GH activate the GHR and causes an increase in IGF1 serum levels	Adding GH antagonist
Autocrine loops in the tumor	Expression of the receptor and ligand by the tumor	IGF1 inhibitors not effective, biomarker studies necessary to select patient who does not benefit from treatment
	Expression of the receptor by the tumor and the ligands by stroma	
Other pathways mutated	Other drivers likeEGFR) or secondary mutations (PI3K or PTEN)	Combination therapy [39]
		IGF1 inhibitors not effective, biomarker studies necessary to select patient who does not benefit from treatment

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
