# Peer review of "Unraveling the Resistance of IGF-Pathway Inhibition in Ewing Sarcoma"

_cancers, 2020, doi:10.3390/cancers12123568_

Round 1
Reviewer 1 Report
In the current form, the manuscript is few informative compared to the abundant literature in the field.
Author Response
Dear reviewer,
We thank you for your time and effort. In accordance with comments from reviewer 3 we have changed the manuscript to be more comprehensive. Extra information has been added to the review, particularly in lines 87-92, 118-126 and 272-277. In addition to this, some grammar adjustments have been made. We hope that by adding this extra information and changing the wording in places, the reviewer will find the new version informative.
Reviewer 2 Report
No further comments.
Author Response
Dear reviewer,
We thank you for your time and effort. In accordance with comments from other reviewers we have improved the manuscript and added some additional information, particularly in lines 87-92, 118-126 and 271-276. In addition to this, some grammar adjustments have been made.
Reviewer 3 Report
Reviewer’s comments
Major revision required
Although the authors have made a good effort to try and address the concerns I raised in the first review I still believe major revision is required before acceptance for publication. Important - please address comments 5, 7, 8, 11, 13, 14 and 15.
Specific comments in chronological order.
Comment 1: Suggest you remove breast cancer from your key words
Comment 2: Is Figure 1 duplicated?
Comment 3: Some inconsistencies in Table 1 in the patient column, i.e. De Bono 125 patients?? And…. What kind of tumors? 10 Japanese patients with solid tumours… what kind of solid tumors?
Comment 4: Tables are comprehensive but suggest reformatting before publication
Comment 5: IGF1 section, needs to be expanded and more informative. The statement, ‘We hypothesized that low levels of IGF1 reflect an endocrine adaptation to severe disease (41)’ please enlarge and explain, insufficient information, i.e. on what basis is the hypothesis?
Comment 6: Lines 104-105. ‘It activates proliferation and anti-apoptotic pathways and seems important in cancer development.’ The statement is ambiguous, a nondescript statement which needs references at best and an explanation on what ‘seems important’ means in this context.
Comment 7: Section starting 111. This section again is very weak. At the very least mention Robert Baxter’s publications. Dr Baxter is a world leading expert in IGFBPs.
Comment 8: Lines 113-114 “This receptor is considered a tumor suppressor as both ligand and receptor are internalized and degraded after binding (49, 50). This statement is not the definition of a tumor suppressor? Please rephrase.
Comment 9: Lines 161 and 167 – check grammar.
Comment 10: Lines 164-165. Line should end inhibitor types.
Comment 11: Preliminary preclinical data. I still think you should rethink how you incorporate raw data in a review. Personally I think you should incorporate the data into the next section to support the hypothesis you are raising rather than a stand-alone entity as preliminary data.
Comment 12: Line 367, Cell growth is 2 words.
Comment 13: Line 367, Preliminary results… - maybe change to ‘our results suggest or support…’. What comes across currently in the review is you are trying to publish unsubstantiated results rather than a well conducted experiment.
Comment 14: Again, line 367 – Preliminary data – suggests the authors are trying to publish data that is incomplete. If the authors are confident of their experiments then it should read, 'our experimental data' or 'our unpublished data supports….'
Comment 15: Supplementary figures. Were these experiments performed in triplicate biological experiments, or, are these experiments ‘one off’ experiments? It is important that you are confident of your supporting results even if you are adding them as supplementary figures. Adding preliminary data is OK for a grant application, but you are using this data to support your hypotheses in this review, in a good journal, I believe you should come across more confidently about your results: be more explicit in how many times you conducted the experiments and perhaps add as supporting data rather than preliminary results which come across as, maybe or maybe not, upheld if you repeat your experiments.
Comment 16: Other – please go through the manuscript carefully for grammatical errors.
Author Response
Dear reviewer,
Thank you for your time, comments and advice. We will go over the comments you had point by point:
Comment 1: Suggest you remove breast cancer from your key words.
We agree as breast cancer is not one of the focusses of this review. The keyword has been removed.
Comment 2: Is Figure 1 duplicated?
The ‘second figure 1’ has been replaced by a higher resolution version. The one you still see has a red bar through it which means it has been removed from the manuscript. It was only still there as a correction in word, but this has been removed.
Comment 3: Some inconsistencies in Table 1 in the patient column, i.e. De Bono 125 patients?? And…. What kind of tumors? 10 Japanese patients with solid tumours… what kind of solid tumors?
With solid tumors we meant that patients with a variety of tumors were admitted to the clinical trials including hepatocellular carcinoma, esophagus carcinoma, colorectal carcinoma, osteosarcoma etc. We have adjusted the description of the table to clear this up as well as added more information about how many Ewing sarcoma patients were in each trial if there were any.
Comment 4: Tables are comprehensive but suggest reformatting before publication
Table 1 has been reformatted. Additionally the table now shows what phase the trials were in, how many patients in the trials were Ewing sarcoma patients.
Comment 5: IGF1 section, needs to be expanded and more informative. The statement, ‘We hypothesized that low levels of IGF1 reflect an endocrine adaptation to severe disease (41)’ please enlarge and explain, insufficient information, i.e. on what basis is the hypothesis?
The IGF1 section has been expanded and more information has been added to better explain why lower IGF1 levels is associated with lower EFS in patients. (lines 87-91)
Comment 6: Lines 104-105. ‘It activates proliferation and anti-apoptotic pathways and seems important in cancer development.’ The statement is ambiguous, a nondescript statement which needs references at best and an explanation on what ‘seems important’ means in this context.
This topic of this line specifically about the IRA receptor. We agree that this was not obvious in the way the sentence was worded and have changed the wording.
Comment 7: Section starting 111. This section again is very weak. At the very least mention Robert Baxter’s publications. Dr Baxter is a world leading expert in IGFBPs.
This section has been expanded with some additional information from a reference by Robert Baxter, as this section could indeed have been more comprehensive.
Comment 8: Lines 113-114 “This receptor is considered a tumor suppressor as both ligand and receptor are internalized and degraded after binding (49, 50). This statement is not the definition of a tumor suppressor? Please rephrase.
This sentence on its own is not comprehensive enough. Extra information has been added to better explain why the IGF2R is considered a tumor suppressor. (Line 110-111)
Comment 9: Lines 161 and 167 – check grammar.
The sentences in question have been rephased. (Lines 158-159 and 165)
Comment 10: Lines 164-165. Line should end inhibitor types.
The wording of this sentence has been changed for increased clarity in accordance with the suggestion of the reviewer. (lines 167-18)
Comment 11: Preliminary preclinical data. I still think you should rethink how you incorporate raw data in a review. Personally I think you should incorporate the data into the next section to support the hypothesis you are raising rather than a stand-alone entity as preliminary data.
We think it is indeed better to integrate the preclinical data in the next section. The preclinical data section itself has been removed and rewritten into the next section (lines 205-206 and 230-237)
Comment 12: Line 367, Cell growth is 2 words.
It indeed is, this has been adjusted.
Comment 13: Line 367, Preliminary results… - maybe change to ‘our results suggest or support…’. What comes across currently in the review is you are trying to publish unsubstantiated results rather than a well conducted experiment.
The word ‘preliminary’ indeed infers that we might be unsure about our data. This has been removed in multiple places from the manuscript.
Comment 14: Again, line 367 – Preliminary data – suggests the authors are trying to publish data that is incomplete. If the authors are confident of their experiments then it should read, 'our experimental data' or 'our unpublished data supports….'
As said for comment 13, the word ‘preliminary’ has been removed from the manuscript.
Comment 15: Supplementary figures. Were these experiments performed in triplicate biological experiments, or, are these experiments ‘one off’ experiments? It is important that you are confident of your supporting results even if you are adding them as supplementary figures. Adding preliminary data is OK for a grant application, but you are using this data to support your hypotheses in this review, in a good journal, I believe you should come across more confidently about your results: be more explicit in how many times you conducted the experiments and perhaps add as supporting data rather than preliminary results which come across as, maybe or maybe not, upheld if you repeat your experiments.
The data showed here are one of experiments, however we are confident that the data shown is representative. For example, we performed the experiment shown in figure 1 in multiple different cell lines and they all showed the same results. The word “preliminary” has been removed from the manuscript as we trust our data.
Comment 16: Other – please go through the manuscript carefully for grammatical errors.
The manuscript has been read over and grammar has been adjusted in several places.
Once again, thank you for your time.
Round 2
Reviewer 1 Report
Despite some few changes, I don't believe the manuscript is suitable for publication.
The review is not highly detailed, furthermore the focus on Ewing sarcoma is not completely addressed, as several general mechanisms are described.
The effort is not enough in my opinion.
Author Response
We thank the reviewer for the comments.
We share the opinion that the manuscript is not entirely focussed on Ewing sarcoma, but we believe that the information from other tumor entities and pathways are relevant to this sarcoma entity as well.
Additionally, we have followed the comments of the other reviewers to further improve the manuscript. We again thank the reviewer for their time.
Reviewer 3 Report
The authors have addressed all my concerns and I recommend acceptance of the manuscript with minor alterations.
1) With the addition of new citations, please make sure the citations are numbered in the references correctly.
2) For aesthetic purpose only maybe the authors may consider reformatting the tables (too many lines which are distracting and take away from the value of the table). This is only an opinion and does not deter from the content.
3) Line 310, supplementary figure? Should this be figures or please specify which figure?
Author Response
We thank the reviewer for their comments and have made some additional changes to the manuscript as outlined below.
Comment 1) With the addition of new citations, please make sure the citations are numbered in the references correctly.
We have gone over all the citations, adjusted the numbering, and put all the references in the right order.
Comment 2) For aesthetic purpose only maybe the authors may consider reformatting the tables (too many lines which are distracting and take away from the value of the table). This is only an opinion and does not deter from the content.
The table has been reformatted to look better aesthetically without changing any of the information in the table.
3) Line 310, supplementary figure? Should this be figures or please specify which figure?
Indeed the figure number which this line referred to was missing, this has been rectified. (now line 287)
Additionally, some minor spelling and punctuation changes have been made.
We thank the reviewer for their time, effort and suggestions.
This manuscript is a resubmission of an earlier submission. The following is a list of the peer review reports and author responses from that submission.
Round 1
Reviewer 1 Report
In this review article, the Authors describe the most recent literature on the mechanisms of resistance to anti-IGF1 therapy in cancer, with a particular focus on Ewing sarcoma.
The IGF/Insulin pathway is a complex system involved in the regulation of a number of biological events in both physiologic and pathologic conditions, including cancer. However, targeting the IGF1/IGF1R pathway to get an anti-tumor effect has given disappointing results in clinical trials. In this review, the mechanisms potentially involved in resistance have been described. The Authors suggest the insulin may cause resistance to IGF1 inhibition, as it stimulates proliferation and survival in Ewing cell lines. I believe the data presented aren't strong enough to support the conclusions of the Authors. First, this manuscript has been submitted as a review manuscript, therefore it should contain and discuss only published, solid data. The data presented in Figures 2-4 are somehow questionable: 1) the concentration of insulin used is quite high compared with the physiologic concentration; 2) additional tools other than cell debris should be used to evaluate cell death/viability.
The manuscript should be modified to discuss only already published studies.
Author Response
Deer reviewer,
We read your comments and agree that we need to choose between research article or review. We decided to go for a review article, leaving a few sentences and one small alinea about the data and observations we have made, this also means a new abstract has been written and the discussion and conclusion have been slightly changed. I will now address all the comments point by point.
Point 1. As stated, we opted to restructure the manuscript into a review. We left a single Alinea of the data in it because we believe it supports the points made in the review and other articles as well. However the main structure is a review and if the reviewer disagrees with this, we can take this bit of data out as well. I hope this answers your comments.
Reviewer 2 Report
The review article entitled ”Unraveling the resistance of IGF-pathway inhibition 2 in Ewing sarcoma and Breast cancer”is well-written. In addition, preclinical data further support the authors' hypothesis that insulin causes resistance to IGF1 inhibition. I have only two minor comments.
1. The resolution of some figures can be improved. Such as Figure 1, Figure 2, and Figure 4. The text labels in these figures is blur in the combined PDF file.
2. Six cell lines were used in this study. Please provide the source of these cell lines in Materials and Methods.
Author Response
Dear reviewer
We read all the reviewer comments and agree that we need to choose between research article or review. We decided to go for a review article, leaving a few sentences and one small alinea about the data and observations we have made, this also means a new abstract has been written and the discussion and conclusion have been slightly changed. I will now address your comments point by point.
Comment 1
Due to the shift to a review, figure 2 and 3 have been taken out of the manuscript. Figure 1 has been replaced with a higher resolution figure.
Comment 2
Due to the shift to a review, this aspect of the manuscript has been removed and seems therefore no longer necessary.
I hope this answers your comments
Reviewer 3 Report
Reviewer comments - Title: Unraveling the resistance of IGF-pathway inhibition in Ewing sarcoma and Breast cancer
This paper is presented as a narrative review on ‘resistance of IGF-pathway inhibition in Ewing sarcoma and Breast cancer’ with original data included from the authors.
Comments
Comment 1: I hate to reject papers outright as I understand the volume of work that goes into every paper. However I am not sure what to make of this paper in its present format. It is neither a good comprehensive review of the literature, nor a good original research article and therefore in its present form I would be reluctant to accept it for publication. It has the potential to be an interesting article, and in parts shows insight, it would be much better if it the manuscript was focused on ewing sarcoma, the main thrust of the article. Breast cancer is mentioned in the title and then barely mentioned throughout the text: breast cancer does not even get a mention in the abstract and has a cursory mention in the introduction, although there are citations to breast cancer articles and clinical trials. Alternatively, the paper would be more presentable if the authors did a comprehensive review on IGF-pathway inhibition in cancer, or an updated review on the resistance of IGF-pathway inhibition in cancer rather than piecemealed narrative review (not quite sure what the authors’ definition is of a narrative review) together with original data that probably would need more rigorous experimentation and clarity to be acceptable in a stand-alone article.
Note Table 1 – In the body of the text (introduction) there is a reference to numerous clinical trials in breast cancer in Table 1, however there is no mention of breast cancer in Table 1. Solid tumors – yes, but these could be any type of solid tumor.
Comment 2: All the figures need titles and more descriptive legends.
Comment 3: Some sections lack any depth: cursory explanations or statements.
Comment 4: With reference to the experimentation there need to more rigor in the experimentations (lacking adequate controls) and also in the explanations in the text: i.e. more clarity is needed in the experimentation.
Example 1: 2.1 Disruption of negative feedback (line 246)??? Please explain?
line 247 – ‘One of the proposed resistance mechanisms in the clinic is that IGF1R inhibition causes an increase in GH and subsequently IGF1 due to disruption of negative feedback’. Please provide reference here.
I am not sure how this experiment fits well with the assumption (line 247 and title) and what the authors mean by negative feedback mechanism in this instance. It is well documented that if you dramatically reduce FBS in the culture medium cells are very unhappy. Also it is well known that insulin is a stimulant in many cancer cells. I am confused with the clarity and significance of the message, Disruption of negative feedback?
Example 2: Figure 4 – no error bars or stats, how many times was the experiment conducted? Too little evidence to say the difference in results was due to mutational status per se.
Figure 5, Where are the controls for this experiment (no treatment)?
Comment 5: In the supplementary data you need to identify/reference were the cell lines came from, i.e. the cell bank ATCC?
Overall - There are many examples of lack if clarity and rigor in experimentation which makes it difficult to accept this article for publication in its present form.
Author Response
Dear reviewer
We read all the reviewer comments and agree that we need to choose between research article or review. We decided to go for a review article, leaving a few sentences and one small alinea about the data and observations we have made, this also means a new abstract has been written and the discussion and conclusion have been slightly changed. I will now address your comments point by point.
omment 1
We agree and have changed the title to not contain breast cancer anymore because the main text focusses more on Ewing sarcoma indeed. Furthermore the paper has been restructured as a review. We left a small bit of data, as we felt it supports some of the points made in the review.
Comment 2
Most figures have been removed due to the data being removed. A title has been added to figure 1 and a more descriptive legend has been added. (lines 86-87)
Comment 3
In multiple places in the manuscript some language has been adjusted and some information has been added. (167-169)(366-376)
Comment 4
Due to the shift to a review structure, most of the data has been removed from the manuscript. These lines have therefore also been removed.
Comment 5
The cell lines used in the manuscript were in reference to the experiments which have been mostly removed from the article. Therefore this information seems no longer necessary to add to the manuscript.